# Modulating Gene Expression within a Microbiome Based on Computational Models

**DOI:** 10.3390/biology11091301

**Published:** 2022-08-31

**Authors:** Liyam Chitayat Levi, Ido Rippin, Moran Ben Tulila, Rotem Galron, Tamir Tuller

**Affiliations:** 1Department of Biomedical Engineering, Tel-Aviv University, Tel Aviv 997801, Israel; 2Sackler Faculty of Medicine, Tel-Aviv University, Tel Aviv 997801, Israel; 3The Sagol School of Neuroscience, Tel-Aviv University, Tel Aviv 997801, Israel

**Keywords:** population genomics, microbiome engineering, gene expression, horizontal gene transfer, evolutionary systems biology, synthetic biology

## Abstract

**Simple Summary:**

Development of computational biology methodologies has provided comprehensive understanding of the complexity of microbiomes, and the extensive ways in which they influence their environment. This has awakened a new research goal, aiming to not only understand the mechanisms in which microbiomes function, but to actively modulate and engineer them for various purposes. However, current microbiome engineering techniques are usually manually tailored for a specific system and neglect the different interactions between the new genetic information and the bacterial population, turning a blind eye to processes such as horizontal gene transfer, mutations, and other genetic alterations. In this work, we developed a generic computational method to automatically tune the expression of heterologous genes within a microbiome according to given preferences, to allow the functionality of the engineering process to propagate in longer periods of time. This goal was achieved by treating each part of the gene individually and considering long term fitness effects on the environment, providing computational and experimental evidence for this approach.

**Abstract:**

Recent research in the field of bioinformatics and molecular biology has revealed the immense complexity and uniqueness of microbiomes, while also showcasing the impact of the symbiosis between a microbiome and its host or environment. A core property influencing this process is horizontal gene transfer between members of the bacterial community used to maintain genetic variation. The essential effect of this mechanism is the exposure of genetic information to a wide array of members of the community, creating an additional “layer” of information in the microbiome named the “plasmidome”. From an engineering perspective, introduction of genetic information to an environment must be facilitated into chosen species which will be able to carry out the desired effect instead of competing and inhibiting it. Moreover, this process of information transfer imposes concerns for the biosafety of genetic engineering of microbiomes as exposure of genetic information into unwanted hosts can have unprecedented ecological impacts. Current technologies are usually experimentally developed for a specific host/environment, and only deal with the transformation process itself at best, ignoring the impact of horizontal gene transfer and gene-microbiome interactions that occur over larger periods of time in uncontrolled environments. The goal of this research was to design new microbiome-specific versions of engineered genetic information, providing an additional layer of compatibility to existing engineering techniques. The engineering framework is entirely computational and is agnostic to the selected microbiome or gene by reducing the problem into the following set up: microbiome species can be defined as wanted or unwanted hosts of the modification. Then, every element related to gene expression (e.g., promoters, coding regions, etc.) and regulation is individually examined and engineered by novel algorithms to provide the defined expression preferences. Additionally, the synergistic effect of the combination of engineered gene blocks facilitates robustness to random mutations that might occur over time. This method has been validated using both computational and experimental tools, stemming from the research done in the iGEM 2021 competition, by the TAU group.

## 1. Introduction

The word “microbiome” refers to various types of microorganisms that exist together in a certain environment. Almost every system, natural or synthetic, is colonized by a distinct and varied population of organisms that engage with one another and with their surroundings on a regular basis. Related early research has revealed that the microbiome of an animal has a significant impact on critical characteristics, such as the lifespan and the fitness of the animal [1]. In recent years, research on the human and animal microbiome has yielded genuinely transformative discoveries, revealing new insights into the crucial role of numerous physiological [2] and even mental [3] aspects the microbiome has on its host, as well as into the interactions between them. Furthermore, investigations into different natural ecosystems have generated promising findings [4]. Several publications have researched and established the tendency of microbiome composition to respond and further modulate environmental changes, marking them as a desirable target for bioengineering, promoting the development of diverse engineering methodologies.

Some strategies that entail the introduction of new or changed bacteria into the environment, for example genomic engineering, and microbiome directed evolution, have been successful in some scenarios [5,6,7,8], mainly over limited periods of time. The underlying reasoning is that transplanted bacteria are adjusted better to their original ecosystem, hence they have a disadvantage while competing for environmental sources in comparison to local bacteria. Other strategies, which have directly targeted the transformation of the vector [9], try to design the natural microorganisms that are already present in the environment. The majority of these techniques were manually created for a specific environment in methods that aren’t easily adapted to new systems once the original system has been developed. The following are some examples: Fecal microbiome transplants have been shown to have positive effects for various bowl diseases such as Crohn’s disease [5,8] and clostridium difficile [10];The use of genetically modified bacteria in soil microbiome improves nitrogen fixation in crops and is a commercially available technology [11];The gut brain axis and connections between the microbiome and mental health have been largely explored, mostly in animal models. The use of fecal matter transplant as treatment has been largely explored [12] although most studies have been conducted in animal models [13].

Bacteria share genetic information through horizontal gene transfer [14] regularly, in many cases the designed process must be delivered exactly to the desired hosts in order to be successful and efficient. Furthermore, because biological systems are inherently unpredictable; introduction of new genetic information into unknown and potentially unwanted hosts might have unforeseen ecological impacts.

Rather than dealing with genetic constructs as they are, this work offers a different perspective over said gene blocks in which each genetic component associated with gene expression is analyzed and artificially transformed. In the human gut microbiome, for example, certain bacteria are symbiotic while some are harmful [15]. In this case, an efficient modeling approach would most likely target a subset of pathogenic bacteria, which can be viewed as the wanted hosts, and create a modification that includes, for instance, a gene that reduces their growth rate. The symbiotic bacteria can be viewed as the unwanted hosts since the modification should maximally avoid expression in it. 

Since horizontal gene transfer (HGT) is such a central property of microbiomes, it naturally dictates the development of microbiome engineering techniques, posing a situation in which, all members of the microbiome are exposed to the plasmidome [16]. Moreover, the process of HGT can either be complete or partial [17]. As a result, in order to maintain selectivity for longer periods of time, each and every element associated with the expression of the engineered gene should promote the expression of the gene selectively, in case the HGT process is incomplete.

Furthermore, this technique can be applied even with relatively limited genomic data and utilize different levels of characterization that might be collected for a microbiome. The present approach works with annotated genomes, but it might also work with metagenomically assembled genomes. Finally, this approach is intended to alter the microbiome over an extended time duration. Each genetic component is analyzed and handled separately, therefore the genetic information is relatively resistant to environmental harm, as seen in Figure 1. The design approach examines the effect of the genetic modification on the chosen host’s fitness and regulates the load exerted on it accordingly.

## 2. Methods

Genes are expressed using a combination of different mechanisms and regulations. The process of evolution is driven by the occurrence of small and random changes, due to the inherent inaccuracy of biological machinery. These changes are then subjected to the power of natural selection, dictating which ones will be passed to future generations. As they drive the process of differentiation into species and phylogenetic progression, they affect every machinery and component in the cell, including the parts related to gene expression. 

By creating biophysical models for gene expression and regulation processes, the cumulative effect of random alterations is highlighted and emphasized, facilitating the different species-specific preferences of the agents which carry out these processes. In this paper, we will focus on three main steps in this process, starting from the entry of foreign genetic information into the cell, through the process of transcription up until the translation into proteins (see Figure 2). The synergistic effect of engineering all these genetic elements allows the engineered sequence to be expressed differently in different hosts in terms of efficiency and expression rate, and in terms of the fitness burden they impose on their host. 

Fitting genetic elements to a microbiome is defined in this paper in a rather generic manner. Once the gene itself is selected, there are two sub-communities of interest; first is the community of organisms that should be able to express the modification and will be referred to as the “wanted hosts”. The second is the group of species that should not be able to express this modification, named the unwanted hosts. The goal of the optimization process is to increase expression in the set of wanted hosts, while simultaneously decreasing expression of the same sequence in the unwanted host, considering the fitness effect on both sub communities. 

These biophysical algorithms are then used to accumulate and combine the small differences occurring between different species, amplifying their collective effect while remaining within the limitations of noisiness of metagenomic data. The architecture and engineering process facilitated by the algorithms, along with the use of additional databases, helps fill in the missing details and characterization automatically. However, in order to utilize the full potential of the microbiome specific collected data, additional data can be submitted and analyzed flexibly.

Algorithms exhibited in Figure 2 were specifically designed to leverage the degrees of freedom in the appropriate genetic elements, specifically focusing on those that are easily derived from conventional characterization of the microbiome, meaning through metagenomic techniques. However, they are also designed to accommodate additional information gathered on the community or some of its’ species, to ensure full utilization of the existing data and the best possible fit to the objective. To sum up, the defined objective and consecutive designing process are essentially used to create a maintainable distinguishment between the set of wanted and unwanted hosts, aiming to differentiate expression levels despite constantly occurring HGT.

### 2.1. Translation Efficiency

Genes themselves are regulated with multiple elements, but the region that specifically codes for the protein itself is named the coding sequence (CDS) or the open reading frame (ORF), as it is the frame read by the ribosome. One of the most interesting characteristics of the genetic code, aside from its universality, is the innate redundancy present in it, posed by the combinatorial number of available codons and existing amino acids. As described in the previous section, this degree of freedom has been subjected to random changes which compile into general phylogenetically dictated preferences of the translation machinery. This preference of specific synonymous codons over others for the same amino acid is usually quantified using metrics called codon usage biases (CUB) [18,19].

Since ribosomes are a limiting resource in cells, these so called “synonymous changes” in the ORF may have a significant in vivo effect by influencing the flow of ribosomes on the mRNA sequence (Figure 3) [20], thus modulating translation efficiency and overall fitness. Utilization of CUB for gene expression optimization is commonly referred to as “codon harmonization”. In order to create a novel optimization technique based on the traditional one, there are two main challenges to face:

Codon harmonization is used to increase translation efficiency of a sequence for a specific organism, meaning in the context of a single proteome, considering a single set of gene expression machinery [19,21,22]. For the objective of this engineering process, the preferences of the entire microbiome must be taken into account (more specifically, the organisms deemed as relevant for the engineering process);Instead of solely optimizing expression, facilitation of increasing expression for the set of wanted hosts should be coupled to impairment of expression in the unwanted hosts.

The degree of complementarity between the sequence and its’ corresponding microbiome should be carefully modulated considering these concerns. 

**Data for CUB calculation**: various techniques and points of view can be used in order to emphasize different aspects of the CUB, usually limited by available data. This tool is equipped to deal with the two main methods to do so; the first looks at the frequencies of different codons in different types of genes, assuming highly expressed genes are likely to have optimal codons, called the codon adaptation index CAI [18]. Secondly, the tRNA adaptation index (tAI) [23] takes into account the supply of different tRNAs and the demand of codon frequencies, considering the interaction strength between the codon and the anticodon. Both options are available and should be used in accordance with the available data regarding the selected microbiome. 

Once the CUB scores of each organism are calculated, the most important features are the mean (μ) and standard deviation (σ), calculated for every considered proteome (meaning all wanted and unwanted hosts).

*Codon adaptation index (CAI):* the underlying assumption is that highly expressed genes have a higher selective pressure to be optimally expressed, thus they are more likely to be consistent of codons that are translated efficiently. In other words, the penalty of having a non-optimal codon out of the synonymous options is much higher in terms of fitness in highly expressed genes compared to lowly expressed genes [18]. According to this understanding, a set of highly expressed genes is obtained and defined as the reference set, either by measuring the protein or mRNA expression levels, or by choosing a set of genes that are known to be highly expressed by homology (such as ribosomal proteins). 

Each codon has a usage score $w_{i}$, named the reference set usage score (RSCU) [18], that is calculated based on a normalized version of the frequency of each synonymous codon $x_{i}$ for amino acid x.
(1)$$\;\;\;w_{i}=\frac{\;RSCU_{codon}}{RSCU_{max}}=\frac{\frac{x_{J}}{\hat{x}}}{\frac{max_{1\leq j\leq d}x_{j}}{\hat{x}}}=\frac{x_{i}}{max_{1\leq j\leq d}x_{j}}\;\;\;\;$$

In order to translate the scores from single codon scores to scoring a gene with L codons, the geometric mean calculation is applied to all RSCU scores of the codons of a gene of interest (GOI).
(2)$$\;\;\;CAI_{seq}={\left(\overset{L}{\underset{i=1}{\prod }}w_{i}\right)}^{\frac{1}{L}}=exp\left(\frac{1}{L}\overset{L}{\underset{i=1}{\sum }}ln\left(w_{i}\right)\right)\;\;\;\;\;\;\;\;\;\;$$

*tRNA adaptation index (tAI):* CAI is calculated from an evolutionary perspective, highlighting the selective pressure effects on fitness. The tAI measure takes a different approach, aiming to capture the effect of interaction strengths between components of the ribosome, and the supply of said reaction components, highlighting factors related to the physiochemical state of the cell. Each synonymous codon is characterized considering the codon-anticodon noncovalent bond strength, and the corresponding abundance of the recognizing tRNA, as each codon can be recognized by numerous tRNA molecules by wobble interactions. 

In order to determine the abundance of different tRNAs, the most optimal measurement would be to measure the expression of the tRNA molecules themselves. However, while this method might work very well for gene expression, tRNA molecules are highly modified RNA sequences and are also very similar to each other, making sequencing outputs inaccurate. The selected measure for this purpose is the tGCN, tRNA genomic copy number of the different tRNAs, using the correlation between the copy number of the molecule and its contribution to the tRNA pool [23]. 

The reasoning behind this is similar to the reasoning of the CAI measurements. Citing fitness considerations, if a tRNA that is highly used has a gene duplication, it will allow the cells to produce more of said tRNA and reduce the expression burden of the specific gene. The calculation of translation efficiency measure $W_{i}$ of codon i depends on the interaction strength $s_{ij}$ and tGCN of all $n_{i}$ recognizing anticodons:(3)$$\;\;\;w_{i}=\overset{n_{i}}{\underset{j=1}{\sum }}\left(1-s_{ij}\right)*tGCN_{ij}\;\;\;\;\;\;\;\;\;\;$$

Normalization of each score and calculation of the final score of each gene are done in the same manner as in the CAI score:(4)$$\;\;tAI_{seq}={\left(\overset{L}{\underset{i=1}{\prod }}w_{i}\right)}^{\frac{1}{L}}=exp\left(\frac{1}{L}\overset{l}{\underset{i=1}{\sum }}ln\left(w_{i}\right)\right)\;\;$$

**Whole microbiome optimization:** as mentioned, the main goal of this step is to maintain optimization for multiple organisms while doing so in a selective manner. In order to accomplish those requirements two designing techniques were generated. In the first method the selected model examines the total effect of synonymous changes on the whole microbiome by comparing them to each proteome they will integrate into, named the “proteome relative method”. The second method, called the “individual amino acid method”, calculates and assigns a loss to each codon, without considering the relativity of the scores to the innate score of the microbiome. While the first approach is likely to have a higher degree of complementarity to the microbiome, it creates dependencies between the different amino acids as the protein is considered as a whole, instead of examination of each amino acid individually like in the second method. As such, optimization using the proteome relative method is done using a greedy hill climbing algorithm, which can converge into a local minimum and is more computationally intensive compared to the optimization of individual amino acids. 

*Proteome-relative method:* The effect of a quantitative change in the CUB score of a heterologous gene is relative to the endogenous CUB scores of the proteins in the environment- if the CUB scores of the proteome of a species have a wider distribution and a larger standard deviation, a small change in the CUB of the engineered gene might be less significant. 

Due to the number of constraints and considerations, the algorithm was designed to function greedily in an iterative manner. The iterations were constructed to modify the selected DNA sequence (X) as following (Figure 4):The neighborhood of modified sequences during the iteration are sequences based on X which have all their synonymous codons for a specific amino acid s changed to a specific codon $s_{i}$, so that $X^{\prime }=X\left[s\to s_{i}\right]$;For each modified sequence, the CUB score in every tested organism A is calculated. In order to compare the scores of different proteomes, each is “normalized” by quantifying the number of standard deviations that differ the CUB of the GOI from the average score of the said proteome:(5)$$\;\;\;dist\left(a\right)=\frac{\left(CUB_{a}\left(X^{\prime }\right)-\mu_{a}\right)}{\sigma_{a}}$$


In order to take both optimization and deoptimization for the wanted (A) and unwanted (B) hosts (respectively) into account, the following optimization score is calculated for every engineered sequence X′ in the neighborhood:(6)$$\;\;f\left(X^{\prime }\right)=\alpha \cdot mean_{a\in A}\left(dist\left(a\right)\right)-mean_{b\in B}\left(dist\left(b\right)\right)=\frac{\sum_{a\in A}\frac{\left(CUB_{a}\left(X^{\prime }\right)-\mu_{a}\right)}{\sigma_{a}}}{\left|A\right|}-\frac{\sum_{b\in B}\frac{\left(CUB_{b}\left(X^{\prime }\right)-\mu_{b}\right)}{\sigma_{b}}}{\left|B\right|}$$

The sequence with the most significant optimization score is considered as the template for the following iteration. Termination conditions include hitting a (local) maximum or exceeding the defined number of iterations allowed.

*Individual amino acid method*: Along with the CAI and tAI codon usage bias measurements described in Section 2.1, an additional CUB measurement was added, called typical decoding rate (TDR) [24]. TDR utilizes ribo-seq data, which is a type of RNA-seq in which only the parts of sequences that were covered by ribosomes are sequenced, providing a snapshot of the ribosome placement in the cell at a given moment. Based on the ribo-seq the typical decoding rate of each of the codons is estimated.

Two optimization strategies were used to select the codon that will facilitate the highest expression levels for *B. subtilis* balanced by the lowest possible expression levels for *E. coli*. The weights $w_{i}$ for each score can be calculated with any CUB measurement.

Ratio score (R):(7)$$R_{i}=\sum_{a\in A}\frac{w_{i}}{max\left(\left\{w_{i}\right\}\right)}-\sum_{b\in B}\frac{w_{i}}{max\left(\left\{w_{i}\right\}\right)}\;$$

Difference score (D):(8)$$\;\;\;D_{i}=\sum_{a\in A}\left(1-w_{i}+\max(\left\{w_{i}\right\}\right)+\sum_{b\in B}w_{i}-\;\max\left(\left\{w_{i}\right\}\right)$$

The codon with the highest score will be chosen upon synonymous options to encode for its corresponding amino acids. Abbreviations formatted as “CUB scoring method”- optimization score. CAI results were unique since the ratio and difference optimized sequences were identical.

**Result evaluation**: a novel evaluation score is defined as the average distance between the cluster of wanted hosts and the cluster of unwanted hosts for an additional score, comparing the normalized changes between the initial and engineered sequence. The optimization score for each organism is defined as:(9)$$\;\;\;Score_{a}=\frac{CUB_{a}\left(X’\right)-CUB_{a}\left(X\right)}{\sigma_{a}}$$

A positive optimization score means that the sequence was optimized compared to the non-engineered version, thus for wanted hosts the results should be as positive as possible and for unwanted hosts they should be negative. The formula for the final optimization index:(10)$$\;Opt.\;index=\;\sum_{a\epsilon A}\frac{score_{a}}{\left|A\right|}-\sum_{b\epsilon B}\frac{score_{b}}{\left|B\right|}=\sum_{a\in A}\frac{CUB_{a}\left(X’\right)-CUB_{a}\left(X\right)}{\sigma_{a}\cdot \left|A\right|}-\sum_{b\in B}\frac{CUB_{b}\left(X’\right)-CUB_{b}\left(X\right)}{\sigma_{b}\cdot \left|B\right|}\;\;$$

As shown for the scores themselves, a higher index indicates that translation is more efficient in the wanted hosts compared to the unwanted hosts, while a lower score indicates the opposite. 

### 2.2. Transcription Optimization

Gene transcription is initialized in prokaryotes by the recognition of promoter sequences, which are found upstream to a gene, and the recruitment of TFs to allow RNA polymerase to initiate transcription. The core promoters are defined as the exact segment to which the sigma factor in bacterial RNA-polymerase binds [25]. While core promoters are quite universal, upstream regions contain additional sites that are recognized by TFs. Different TFs, utilized by different organisms, recognize different sets of genomic sequences known as “motifs”. By characterizing motifs that are specifically recognized by wanted and unwanted hosts’ cellular machinery, the transcription module estimates which promoters will promote transcription initiation only in the group of wanted hosts within a microbiome. 

For the purpose of this study, promoter sequences were defined as the first 200 bp upstream to the ORF and intergenic sequences as all sequences on the same strand that neither belong to the ORF nor to the promoter sequences (Figure 5A).

**Motif discovery:** A Position-Specific Scoring Matrix (PSSM) is usually used to represent sequence motifs, as nucleotides can vary in different positions along the sequence. A PSSM of size 4 × L contains the probability of each nucleotide to appear in each position of a motif of length L. PSSM probabilities are calculated assuming motif sites are independent one from another and neglecting insertions or deletions in the motif sequence.

The STREME (Sensitive, Thorough, Rapid, Enriched Motif Elicitation) software tool was used to search for enriched motifs in primary set when compared to a set of control sequences. STREME uses hidden Markov model (HMM) to scan the query sequences for enriched motifs of configured length up to a certain significance threshold [26,27]. In this study, STREME was run with a configuration of third order HMM, motifs’ length of 6–20 bp and a *p*-value of 0.05. Two sets of enriched motifs related to transcription were searched (Figure 5B).

*Transcription enhancing motifs*: to ensure a motif is related to transcription activation in wanted hosts, motifs were searched from the third most highly expressed (inferred from expression data or CUB measurements) promoters of each wanted host with the promoter sequences defined as the primary input and the intergenic sequences as the control. Motifs discovered in this run configuration are enriched in sequences associated with gene expression, which likely indicates their desirable regulatory role.

*Transcription inhibiting anti-motifs:* to verify a motif will not promote transcription in unwanted hosts, motifs were searched for each unwanted host with the intergenic sequences defined as the primary input and the promoter sequences of the third most highly expressed genes as the control. The discovered motifs, termed “anti-motifs”, are common in sequences that are not associated with gene expression, which likely indicates their transcription suppressing attribute.

For an input of n wanted hosts and m unwanted hosts a total of n+m sets of motifs are created.

**Single motif set construction:** Let A be the set of wanted hosts and B the set of unwanted hosts. $\forall a\in A\;S_{a}$ is the set of transcription enhancing motifs for host a and $\forall b\in B\;S_{b}$ is the set of transcription anti-motifs for host b. A final set F of motifs is constructed according to the following steps (Figure 5C): 

*Calculating motif similarity thresholds:* The measurement used to quantitate motif similarity is spearman correlation, which is calculated between a pair of PSSMs. In order to determine the basic amount of similarity between motifs to consider, a different threshold is calculated for each organism. ∀h∈A∪B:$PSSM_{h}\;$ is a set of 100 random PSSMs with lengths 6–20 bp;$\forall m\in \;S_{h\;},\;\forall m\prime \in \;PSSM_{h}$ spearman correlation corr(m, m′)  between m and m’ is calculated. Let $corr_{h}{}_{\;}=\left\{corr\left(m,\;m\prime \right)\;|\;m\in \;S_{h\;},\;m\prime \in \;PSSM_{h}\right\}$;Let $P_{X}\left(corr_{h}{}_{\;}\right)\;$ be the  X-percentile of the spearman correlation values. The motif similarity threshold used for host h is defined as $D_{h=}P_{X}\left(corr_{h}{}_{\;}\right).\;$ For the purpose of this study, X=95 was set to determine motif similarity threshold for each host.


*Defining an initial motif set:* The initial motif set C is defined as the union of all transcription-enhancing motifs, $C=\cup_{a\in A}S_{a}$. 

*Calculating motif scores:* ∀m∈C two motif scores are calculated.

Motif score for a single organism:(11)$$\;\delta_{h}\left(m\right)=\left\{{\;}_{0\;\;\;otherwise}^{1\;\;\;if\;\exists m’\in S_{h}\;s.t.\;\;corr\left(m,m’\right)\;\geq \;D_{h}}\right\}\;\;\;\;\;\forall h\in A\cup B$$

Aggregated motif score, using a tuning parameter α for calibrating the ratio of modification in wanted hosts and unwanted hosts:(12)$$\;\;\;Score\left(m\right)=\alpha \;\cdot \;\underset{a\in A}{\sum }\delta_{a}\left(m\right)+\left(1-\alpha \right)\;\cdot \underset{b\in B}{\sum }\delta_{b}\left(m\right)$$

*Constructing the final motif set:* Let $Score_{F}=\left\{Score\left(m\right)\;|\;m\in C\right\}$. Let $P^{*}{}_{Y}\left(Score_{F}\right)\;$ be the Y-percentile of the aggregated motif scores. The final motif set F is defined as:(13)$$F=\left\{m\;|\;\;m\in C,\;\;Score\left(m\right)\;\rangle \;P^{*}{}_{Y}\left(Score_{F}\right)\;\;\right\}$$

For the purpose of this study, Y=75 was used to calculate final motif score for each motif. 

**Promoter selection and tailoring:** MAST (Motif Alignment and Search Tool) [27] is used to align the final motif set to the top quartile of wanted hosts’ promoters in terms of gene expression, when gene expression data is available or estimated based on CUB scores calculated from the hosts’ genes. The hosts’ promoters are then ranked based on the Expect Value (E-value) of those alignments, considering the initial significance of the motif and the quality of the alignment (Figure 5D). Top ranked promoters, identified as best candidates to promote transcription initiation exclusively in wanted hosts, are further tailored by individually alternating mismatches between the mapped motifs and the promoters (Figure 5E).

### 2.3. Editing Restriction Site Presence

Restriction enzymes are the first line of defense in the bacterial immune system, they have a specific ability to recognize a nucleotide sequence and digest it, thus protecting bacteria from the effects of foreign DNA entering it. One of the most variable properties of different bacteria, is their array of recognized restriction sites and footprint of restriction enzymes, as demonstrated in Figure 6.

The cleaved product may have different forms, depending on the specific type of restriction enzyme which performed the cleavage action. In some cases, the digestion products have complementary edges that can reattach due to the bacterial DNA repair mechanisms [28]. Therefore, two main factors determine the effectivity of the digestion process: the number recognized restriction sites and the region in which the sites are introduced.

As in other bio-design cases, the topological complexion and scale of effect of the genetic element is directly related to the magnitude of its effect and engineering potential of it. As result, this work focuses solely on the ORF and does not modulate other genetic elements, in fear to disrupt its action as a byproduct. Changes in the ORF sequence have the most predictable outcome, making them the best candidate for this algorithm. 

**Restriction site detection and filtration:** in this preprocessing step, each restriction site is classified as one of the following: sites uniquely recognized by the wanted hosts or unwanted hosts, and sites recognized by both. The goal of this algorithm is to avoid any site present in a wanted host, whether or not it is present in an unwanted host as well, while simultaneously adding sites recognized only by the unwanted hosts without disrupting the sequence of amino acids.

**Insertion of sites**: overlapping sites can obviously not be inserted together, as the insertion of one site disrupts the presence of the other, thus the objective is to specifically introduce sites that maximize the number of unwanted species that can recognize and digest the sequence, as the total number of present sites is also pursued as a secondary goal.

In order to increase the overall probability of digestion as defined, all appearances (current and potential- given synonymous changes) are located along the sequence. The first sites to be incorporated are the conflict free sites. In the overlapping site case, as described in Figure 7 and Figure 8, sites will be prioritized based on how much they increase the number of unwanted hosts that recognize at least one site in the sequence, and if two options have the same rank, they will be chosen based on the overall number of sites recognized by the unwanted hosts, prioritizing hosts with less found sites.

The order in which the conflicting sites are resolved obviously determine the final DNA sequence to some degree as well, due to the greediness of the algorithm. In order to get the most optimal result given the insertion mechanism, the sites are iterated starting from those with the least complicated conflict up to those with the highest degree of complexity and as result, largest degree of freedom in the final choice of the site.

**Avoidance of sites originating from wanted hosts:** The sites from the first and third group should be avoided, and their presence in the engineered sequence should be disrupted and altered using synonymous changes if possible. This algorithm re-writes this requirement as constraints that can be applied to the sequence using the DnaChisel [29] software tool. An important highlight to this method is that the order of these steps is meaningful, as insertion of a restriction site recognized by an unwanted organism can create a new restriction site that might be recognized by a wanted host, reversing the goal of the optimization process.

The Restriction enzyme database (Rebase) is a database of information about two types of enzymes: restriction enzymes and methyltransferases. The characterization of these enzymes details their origin, recognition sites, and other metadata such as the year of discovery or commercial availability. The detailed sites themselves are noted using standard abbreviations to represent sequence ambiguity, and in some cases note the exact digestion pattern and resulting ends. 

The database is constantly updated, as the rate of metagenomic sequencing increases, the fraction of computationally inferred restriction enzymes becomes more prominent (along with reducing rates of biochemical characterization of the sites) [30,31]. 

In version 110 (release date: 28 September 2021), there is a total of 4735 organisms (388 of them labeled as ‘Unidentified bacterium’ which were ignored from future analysis), as all strains of the same species were considered altogether. They contain a total number of 5488 sites, with an average of 4.6 sites per organism. The number of sites highly varies, as the standard deviation is 21.2. 

## 3. In Vitro Methods

**Materials and plasmids:** PCR master mix, *DpnI*, Gibson Assembly kit, PCR cleaning kit, competent *E. coli* and plasmid miniprep kit were purchased from *NEB*. *E. coli k-12, B. subtilis PY79* and AEC804-ECE59-P43-synthRBS-mCherry plasmid were kindly provided by Prof. Avigdor Eldar (Tel-Aviv University, IL, USA, Figure 9). Agarose for DNA electrophoresis, Chloramphenicol, M9 minimal media and 96-well black plates were purchased from *Sigma*. LB and agar were purchased from *BD Difco*, and Ethidium Bromide solution was purchased from hylabs. Modified versions of gene of interest (GOI)and primers were synthesized by *IDT*.

**Solutions:** Bacillus transformation (BT) solution: 80.5 mM dipotassium dihydrate, 38.5 mM potassium dihydrogen phosphate, 3 mM trisodium citrate, 45 µM ferric ammonium citrate, 2% glucose, 0.1% casein hydrolysate, 0.2% potassium glutamate and 10 mM magnesium sulfate in DDW.

*Trace elements solution (x100)*: 123 mM magnesium chloride hexahydrate, 10 mM calcium chloride, 10 mM iron chloride hexahydrate, 1 mM manganese chloride tetrahydrate, 2.4 mM zinc chloride, 0.5 mM copper chloride dihydrate, 0.5 mM cobalt chloride hexahydrate and 0.5 mM sodium molybdate.

*Minimal medium:* 1X M9 solution, 1X trace elements solution, 0.1 mM calcium chloride, 1 mM magnesium sulfate, 0.5% glucose, and chloramphenicol (5 µg/mL).

**Plasmid construction:** software-designed mCherry gene variants were synthesized by IDT and cloned into AEC804-ECE59-P43-synthRBS-mCherry plasmid, to replace the original mCherry gene via Gibson assembly method. Briefly, the original mCherry gene was excluded from the vector by PCR, with primers containing complementary tails to each of the software-designed mCherry variants. PCR products were treated with DpnI to degrade the remains of the original vector and cleaned with PCR cleaning kit. Next, each software-designed mCherry gene was cloned into the vector by Gibson assembly with 1:2 molar ratio (vector: insert) and transformed into competent *E. coli*. Positive colonies were confirmed by colony PCR and sequencing, and the new plasmids were extracted with miniprep kit.

**Bacterial transformation:** all plasmids harboring the modified mCherry genes were separately transformed into competent *E. coli k-12* following the standard protocol, and into *B. subtilis PY79*. For the latter, one bacterial colony was suspended in BT solution (see *solutions*) and grew at 37 °C for 3.5 h. Then, the plasmid was added to the bacterial solution (1 ng/1 uL), and following 3 h incubation, bacteria was spread over pre-warmed agar plates.

**Fluorescence measurement assay:** for each tested mCherry variant, a single colony containing the modified plasmid was grown overnight in LB medium. Then, bacterial suspension was centrifuged and resuspended in PBSx1 twice. Following the second wash, the bacterial suspension was centrifuged again, and the pellet was resuspended in minimal medium (see *solutions*). The bacterial suspension was allowed to grow for 4 h. Then, bacteria were diluted with minimal medium to obtain an OD600 nm of 0.2, loaded into a 96-well plate and grew for 17 h at 37 °C with continuous shaking. Fluorescence (ex/em: 587/610 nm) and bacterial turbidity (at OD600 nm) were measured every 20 min. Each sample was tested in triplicates at three independent experiments. 

**Computational log-phase detection**: growth curves (OD600 nm) were plotted over time, and linearity that represents this phase was detected by sequential removal of the last point in the linear phase. Then, a linear trendline was fitted to the curve, and if the removal of the point increased the slope of the curve, that point was considered not part of the log phase. These iterations were conducted continuously until ⅛ of the graph is left or if two iterations did not change the calculated slope. 

**Statistical analysis:** We calculated *p*-values with a permutation test. Briefly, for every optimization, the three experiments from the same organism were averaged and a difference between *E. coli* and *B. subtilis* was calculated. Then, splitting, averaging, and distance calculations were performed, to assess if the separation between *E. coli* and *B. subtilis* is significant. The P-value is defined as the percent of splits in which the difference between the two is larger than the difference between the original split. 

## 4. Results

Evaluation of the results was carried out through both in silico and in vitro means, each shedding light on different aspects of the engineering process. The in-silico examination tested the resolution of differentiation between wanted and unwanted hosts, and the sensitivity to the community size and complexity, while the in vivo experiment was able to quantify the facilitated change in gene expression, and the effect it had on bacterial fitness.

### 4.1. Translation Efficiency Modeling

The two main characteristics to be tested for this algorithm are the phylogenetic resolution of optimization, and the ability of this engineering approach to scale up.

The selected microbiome for model analysis is a sample of the *A. thaliana* soil microbiome [32], which contained taxonomic lineages and 16S rRNA sequences. The annotated genomes were selected by running the 16S sequence against the BLAST rRNA software [33] (lower threshold for percent identity of the 16S rRNA sequence is 98.5%). As previously mentioned, these algorithms are designed to work with metagenomically assembled genomes in general.

Additionally, the gene used as a target for optimization is the ZorA gene, which serves as a phage resistance gene as part of the Zorya defense system, inferred to be involved with membrane polarization and infected cell death [34]. This gene can be used in a wide array of sub-populations for various purposes, showcasing the flexibility of this framework. 

As a start, the effect of an engineering process on the full microbiome was generated and examined. Figure 10(A.1,A.2) exhibit the optimization starting point, showing CUB scores of each codon in for the examined microbiome. As expected, the initial scores are relatively diverse, showcasing the potential of modulation of this aspect of gene expression. For the particularly tested gene, Figure 10B shows the scores of the native sequence and Figure 10C of the engineered one. Overall, the CUB scores of the optimized sequence are generally regarded to be better compared to the non-engineered version, although the optimization is more substantial for the organisms defined as wanted hosts compared to the unwanted hosts.

Scale up is tested in Figure 11A, where the optimization index is calculated for microbiomes of different sizes, and the most evident result is that for all examined sizes the optimization remains relatively similar and significant, roughly two units of our defined “standard deviation” compared to the initial sequence. Since the degree of optimization is very loosely dependent on the microbiome size, it isn’t hard to believe that this phenomenon will be true for larger and more complex microbiomes as well, and that they will receive a positive optimization index. 

Moreover, the resolution of selectivity than can be achieved by this engineering method, as tested in Figure 11B. For the analysis, every pair of species in the *A. thaliana* soil microbiome were selected and subjected to two examinations- first included calculation of a phylogenetic distance estimate, based on the distance in the 16S sequence alignment between the two species. Next, the algorithm was applied to them (defining one species as wanted and the other as unwanted, reducing the two sets to the size of 1 to investigate the direct effects of this factor). The clear correlation (0.737 spearman correlation) between the optimization index and estimated phylogenetic distance sided with our expectations given the underlying assumptions on which the algorithm was constructed. 

In order to probe the resolution question a bit deeper, the 10% phylogenetically closest pairs were further examined. The mean optimization index is 1.215 ± 0.8, which is a relatively significant optimization in respect to the low degree of phylogenetic diversity presented. In conclusion, these results point out the likelihood of this optimization to be able to effect realistic sizes and microbiome complexities. 

### 4.2. Transcription Optimization

As previously mentioned, promoters have a complex topology, thus the characterization of the effect of any engineering process is less complete compared to other engineered elements. This was taken into account both in the transcription algorithm design and in the analysis, using light selection and modulation in a less direct approach and trying to conserve the innate promoters’ structure as much as possible. 

The evaluation of the designed algorithm was done in two steps; first the ability to differentiate motifs between wanted and unwanted hosts was closely inspected, and only then was the scale up of the algorithm investigated in a similar manner to the translation efficiency model (Section 4.1). 

Examination of the differentiation capacity of the algorithm between wanted and unwanted hosts (Figure 12) was done by examining the ability of the final curated motif set to contrast promoters for two species originating from the A. thaliana microbiome (as curated in Section 4.1). 

Although the two species in Figure 12 originate from the same phylogenetic family, the E-values of the highly expressed endogenous promoters from the wanted host have better match scores with motifs in the final motif set, compared to the unwanted host. This effect can be seen by two different trends-first, the mean and median of the E-values scores of the highly expressed promoters are lower for the wanted host. Second, the exponential trend in the E-values of the wanted host shows the ability of the algorithm to find a selected set of candidates that can serve as selective promoters. This evidence supports the approach implemented in our module, that motifs can indeed be used to construct promoter sequences that will promote transcription only in a group of wanted hosts within a microbiome.

The dataset chosen for examination of the scale up of the algorithm was the MGnify genome dataset [35], which has sets of high quality metagenomically assembled genomes (MAGs) for various environments.

Figure 13A demonstrates the performance of the transcription module for three different microbiomes from MGnify-the human oral microbiome, the cow rumen microbiome, and the marine microbiome. Each of the MGnify sets is built using numerous metagenomic projects and contains high quality MAGs. These MAGs were randomly sampled to examine the effect of the algorithm on small, medium and large microbiome sizes. The phylogenetic richness and quality of the genomes in the samples were not controlled, mimicking the intended usage of the tool in microbiome research.

The overall trend observed in all three microbiomes is the decrease in E-values scores for increasing microbiome sizes. This result is rather counter-intuitive, as larger microbiomes are more complex and thus are expected to be more difficult to optimize. However, since the algorithm uses a single motif set to fit all wanted hosts, in case of a large phylogenetic variance in the wanted hosts group, as may be the case when using randomized samples of species from different phyla, the performance of the algorithm per each host in the group is sub-optimal. When using larger microbiomes, since the dataset used contains species from a finite set of phyla, it is more likely to select wanted hosts with a closer phylogenetic distance, thus improving the fitness of the motifs in the final set for a larger number of wanted hosts.

Additionally, it should be noted that the cow rumen microbiome has overall lower E-value scores for both wanted and unwanted hosts in comparison with the human oral and marine microbiomes, with less differentiation between wanted and unwanted groups. In terms of the number of species (Figure 13B), the human oral microbiome has 452 MAGs, the marine microbiome has 1465 and the cow rumen microbiome has 2686. When observing the number of phyla in each microbiome, the ratio between the number of species (represented as the number of MAGs) and the microbiome size seems to be similar and much larger for the human oral and marine microbiomes compared to the cow rumen microbiome. This observation may indicate that the microbiome richness is the key factor influencing the mentioned difference. 

In a rich microbiome that has phylogenetically distant species, such as the human oral and marine microbiomes, small sub-microbiomes are likely to have higher degree of distinction between wanted and unwanted hosts since the overlap between the groups is relatively small. When the size of the sub-microbiome increases, it is more likely to include wanted and unwanted hosts from overlapping phyla, thus minimizing the differentiation between the groups. When examining microbiomes that are less diverse, such as the cow rumen microbiome, randomly selected species of wanted and unwanted groups are likely to be more similar even for small sub-microbiomes, thus reducing the observed effect of microbiome size, as increasing the sub-microbiome size does not incur a proportional increase in the phylogenetic diversity of the wanted and unwanted hosts which isn’t already captured for smaller sub-microbiomes. 

In conclusion, while the analysis exhibits the ability of the transcription optimization model to differentiate between the group of wanted and unwanted hosts, it also highlights the effect of the phylogenetic richness of the microbiome, adding an additional consideration for the selection process of a sub-microbiome of interest for future applications of the model. 

### 4.3. Editing Restriction Site Presence

In order to capture the full effect of the diversity of known sites and species, the characterized species were used as a pool to select sub microbiomes from, and asses the scale up of our model along with other properties. The optimized sequence is the same one used for ORF optimization, of the ZorA phage resistance gene. 

To examine the performance of our algorithm for different sizes of microbiomes, 10 random microbiomes of the tested sizes were optimized and evaluated.

An important remark to be made prior to more in depth explanations, is to clarify the reason why the engineered sequence even has restriction sites recognized by enzymes present in the wanted group of hosts. A natural assumption could be that this is a byproduct of the insertion of sited from unwanted species, however this is not the case- the last optimization process aims to eliminate the part of this effect that occurs due to that reason. The problem is, restriction sites are innately ambiguous, so in many cases these degrees of freedom are larger than those posed by the redundancy of codons, thus synonymous codon swaps cannot remove the presence of these restriction sites. 

In all cases the new version has a larger amount of restriction sites recognized by unwanted hosts compared to wanted hosts (as seen in Figure 14B), and same goes for the number of species with a corresponding site (as seen in Figure 14A). These two observations are the main desired outcomes for the algorithm- showing that both a larger number of unwanted species will be able to degrade the engineered sequence, and the probability of full degradation (associated with the number of recognized sites) is larger in the unwanted hosts as well (exhibited in Figure 14C). 

Figure 14C gives a spotlight to evaluate the ability of the optimization process to scale up to larger microbiomes, by checking the percent of organisms from each group that have a corresponding site in the engineered sequence for all microbiome sizes. The most evident detail is the lack of a specific trend for both groups; 60% of wanted bacteria that have at least one restriction site in the engineered sequence, compared to 90% of the unwanted hosts, for all sizes.

Additional interesting property was the effect of different ratios of wanted and unwanted species within the microbiome. Analysis of the effect of this characteristic was done by selecting random microbiomes of 30 species, assigning a certain percentile of them as wanted (as the others are assigned as unwanted species), for percentiles ranging from 5–95%. 

Continuing the trend presented in the scale-up examination, for all tested ratios the algorithm was able to create a degree of differentiation between the percent of the group that has a restriction site between the wanted and the unwanted host, remaining largely indifferent to the presented condition. The main conclusion is that not only is the algorithm able to create a degree of distinction between the wanted and unwanted groups of hosts, but that it is able to do so in a way that fully expresses and accommodated the specific customizations and requirements of the engineering process. 

### 4.4. In-Vitro Results

To validate our software’s ability in designing selectively expressed genes, we established a fluorescence assay to monitor the expression levels of the reporter gene mCherry, in two distinct bacteria- *E. coli* and *B. subtilis*. The software generated five variants of the gene’s ORF that were predicted to preferentially express in *B. subtilis* (wanted bacteria), but not in *E. coli* (unwanted bacteria). Each version was cloned into a plasmid (Figure 9), which then transformed into both bacteria. The bacteria allowed to grow separately for 17 h in 96-well plate, while the fluorescence intensity that reflected the expression level of mCherry, and bacterial density were measured for each modified version of mCherry gene (see methods). Alterations in these parameters were compared to the unmodified variant of mCherry gene. 

**Modifying the gene of interest’s ORF arrests the propagation of the deoptimized host:** At first, we determined the effect of each mCherry’s variants on bacterial growth. The measured bacterial density (solution turbidity at 600 nm) was plotted against time (Figure 15A), and logarithmic phase was detected as described in Section 3. Bacterial growth rates were depicted as the slopes of the logarithmic trendline. In *B. subtilis*, all modified variants of mCherry exhibited decreased growth rates compared to ‘wild-type’ mCherry, but maximal bacterial densities were similar (Figure 15B,C). In *E. coli*, however, variants TDR-D, and particularly TAI-D, showed limited growth rates (up to seven-fold change in TAI-D, Figure 15B), as well as reduced maximal bacterial density (Figure 15C). This might be due to ribosomal traffic jams [36] that in turn attenuated overall protein synthesis, and thus restricted bacterial propagation. In order to evaluate the extent of gene selectivity toward *B. subtilis* relative to *E. coli*, growth rates folds (modified mCherry version/unmodified mCherry) of *B. subtilis* were divided by those of *E. coli*. The mCherry variants TDR-D, and more robustly tAI-D, clearly demonstrated selectivity toward *B. subtilis*, with regard to growth rates (Figure 15D).

**Expression levels of the GOI confirm model performance:** Examination of mCherry’s fluorescence intensity, which corresponds to its expression levels in the bacteria, showed that expression of the gene variants correlated with our model predictions. The expression levels over time of all variants were largely amplified in *B. subtilis*, while substantially diminished in *E. coli* (Figure 16A) in comparison to the basal fluorescence of the original variant. These results fit to the chosen optimized and deoptimized bacteria respectively. It was also evidenced by the average maximal fluorescence intensity for all variants, with dominant effect of TAI-D in *B. subtilis* that exhibited three-fold increased and TDR-D in *E. coli* that demonstrated almost four-fold decreased (Figure 16B). To accurately assess the net effect of the software’s modification on the expression of the mCherry gene, we normalized its expression levels with bacterial density, by calculating the ratio of fluorescence intensity per bacterial density. Then, average normalized fluorescence and its fold change relative to the original mCherry variant were determined (Figure 16C). Following normalization, variant TAI-D ranked as the most effective modification in elevating mCherry’s expression in *B. subtilis*, while variant CAI reduced mCherry’s expression to minimum in *E. coli*. Intriguingly, normalized fluorescence of variant TAI-D in *E. coli* was comparable to the unmodified mCherry variant, but also impede bacterial propagation (Figure 16A–C). This imply that gene modification based on tRNA abundance only might not be sufficient for expression selectivity, but it eventually halts bacterial growth. 

Finally, to address the preferential expression degree toward *B. subtilis* relatively to *E. coli*, ratios of normalized fluorescence folds of *B. subtilis* to those of *E. coli* were determined (Figure 16D). In that way, we inferred that GOI versions of CAI and TAI-R were the promising codon usage bias scores.

## 5. Discussion

In this paper, we developed a unique technique for microbiome engineering and demonstrated its computational and experimental capabilities. We were able to establish a complete and automatic strategy for each given consortium of bacteria by effectively integrating the different parts in the gene expression process. We were able to provide a comprehensive view of effects caused by interactions between new genetic information and the existing population in an environment, as well as new inter-microbiome interactions and their corresponding fitness effects, by investing in the adjustment of each genetic element that influences gene expression.

The current model strategy focuses on the three major processes involved in gene expression: cell entry, transcription, and translation. The presence of restriction sites in the bacterial cell is edited, making the plasmid more likely of be digested when introduced into an unwanted host versus a wanted host. The transcription process is then improved by identifying genetic patterns that are most likely connected to TFs that are present specifically in the desired hosts and are involved in transcription initiation. Finally, the translation process involves re-coding the ORF based on translation efficiency modulation, which takes use of the degree of flexibility provided by the genetic code’s redundancy.

We used both in silico and in vitro studies to validate this engineering process. In vitro testing revealed that the optimization method had a positive influence on the expression rate of a chosen protein (Section 3), which was increased for the wanted host (*B. subtilis*) while also decreasing for the unwanted host (*E. coli*). Furthermore, when examining the corresponding fitness effect reflected in the bacterial growth rate, the engineered sequence does not impose a major burden in the wanted host as compared to the initial sequence, the unwanted hosts experienced a significant reduction in fitness. These findings can be enhanced and investigated further by performing a co-culture experiment aimed at studying the dynamics of a population.

The visible decreasing trend in fitness of the unwanted hosts caused by the optimization process may not have a major impact in lab conditions, but the significance of this change is that when dealing with an actual microbiome especially considering the relevant time periods of the modification. There is a stronger evolutionary pressure against the plasmid’s presence in unwanted hosts, therefore further propagating the designed expression differentiation in the microbiome. The in-silico investigation provided interesting insights on each of the computational approaches separately. The significance of the various tests is determined by the scale-up process- from two species to a complete microbiome.

### 5.1. Future Plans

The goal of this paper is to highlight this innovative technique, which suggests that microbiome engineering can be accomplished by adjusting individually genetic components that are related to gene-expression. The following elements should be included in the model in order improve and strengthen the method proposed in this article: **Origin of replication (ORI):** The origin of replication has the greatest impact on the long-term microbiome-plasmid interactions, since it is the genetic factor that stimulates replication of the genetic data. Thus, tailoring the ORI to a specific subset of the microbiome species should considerably increase the efficiency of our technology (as shown in Figure A1 and Figure A3). Learning the structure of the ORI sequence in different creatures, which is exceedingly complicated and changeable, would be the primary obstacle for this model.**Clustered regulatory interspaced repeats (CRISPR) memory:** CRISPR memory is another possibly effective method for distinguishing between distinct species. The modification plasmid is then engineered to contain target sub-sequences that match the unwanted hosts CRISPR system (Figure A2), while excluding target sub-sequences that match the wanted hosts CRISPR system.**miRNA and other mRNA binding RNA genes:** These RNA sequences connect to the mRNA and reduce the rate of protein synthesis. Using calculated effects of miRNA binding sites, sequences identified by RNA genes found in the unwanted hosts will be added to the modification plasmid, while sub-sequences identified by the wanted hosts will be removed, is a similar manner to the use of CRISPR memory.**Translation initiation:** A separate model is needed to represent the processes of connection between the ribosome and the ribosome binding site (RBS) and the translation initiation. This separate model should consider the location of the RBS relatively to the translation start site (TSS), as well as the folding of the mRNA around it (Figure A3).**Multiple optimizations:** The next stage in this suggested method would be to create a set of alternative sequences instead of a single optimized solution. Each sequence in the set will be engineered to have poor expression in the unwanted hosts organisms while having greater expression levels in the wanted hosts organisms (as seen in Figure A4), strengthening the ability of this tool to scale up and operate on more complex microbiomes.**Validations:** The OIL-PCR method by Peter J. Diebold et al. [37] enables quantitative evaluation of bacteria carrying the plasmid, which can be used in order to assess the GOI abundance, we intend to extend this approach to integrate expression as well using the same concepts only relying on fusion-PCR of 16s rRNA genes and the mRNA products of GOI (detailed in Figure A5). This proposed method can be used to determine the overall effectiveness of the engineering technique in a real microbiome.

### 5.2. Applications

The capability of this method to work automatically for a defined microbiome and specified gene is one of its key features. Thus, it might be used in a variety of fields, which have been categorized into three sections:**Bio-sensing:** microbiomes are continually reacting to environment changes, which means they have the ability to detect and respond to them [38]. This method allows tuning of the sensing elements towards certain species of bacteria (shown in Figure A6), which can be used to mark their presence in the environment or receive better signal to noire ratios (SNR) [39,40,41,42,43,44].**Bio production/Bio degradation:** the microbiome’s functional effect on the ecosystem is assisted through biomolecule production and destruction. This natural ability might be used in a variety of situations (some examples are exhibited in Figure A7), including oil bioremediation, some relevant food technology applications, metabolism engineering, etc. [38,45,46,47,48,49,50,51].**Microbiome specific therapy and enhancement:** providing selective expression of heterologous genes within a microbiome can be used in order to repair and add capabilities while improving biosafety and functionality. Soil microbiome therapy [52,53,54] and human microbiome treatment [9,52,53,54,55,56], are two of the most common options in this case.

## Figures and Tables

**Figure 1 biology-11-01301-f001:**
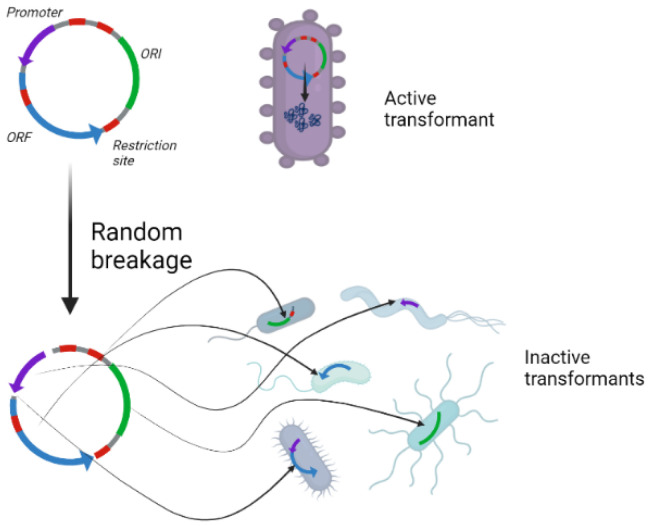
Resistance of the engineering technique to environmental damage. When partial gene transfer, or any other natural alteration occurs, the passed fragments will be inactive, increasing the biosafety of the engineering technique.

**Figure 2 biology-11-01301-f002:**
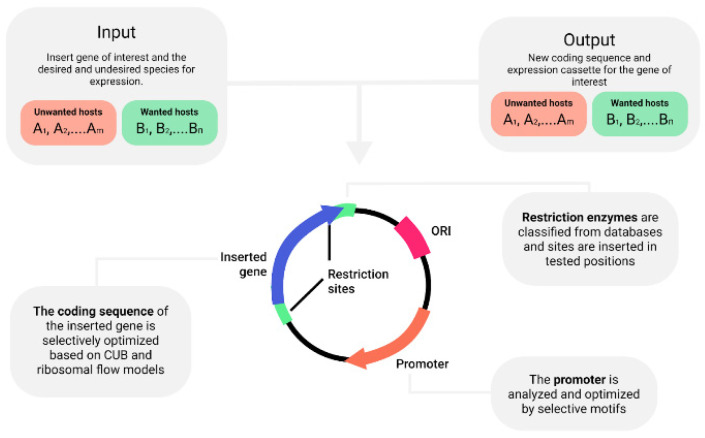
A microbiome specific plasmid and it’s designed elements.

**Figure 3 biology-11-01301-f003:**
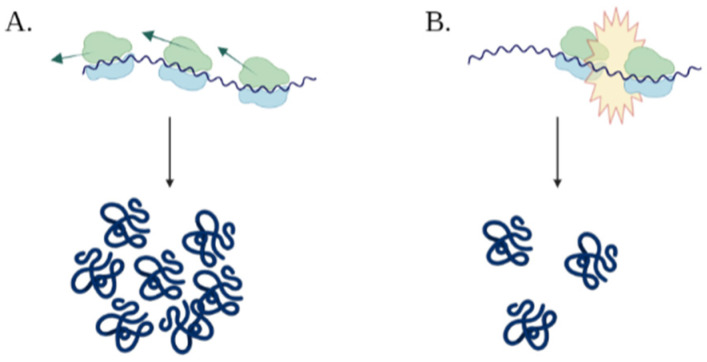
Illustration of the effect of translation modulation. Figure (**A**) describes the effect of translation optimization, causing increased ribosomal flow when efficient codons are used, while figure (**B**) showcases the opposite effect, where ribosomes collide and create traffic jams.

**Figure 4 biology-11-01301-f004:**
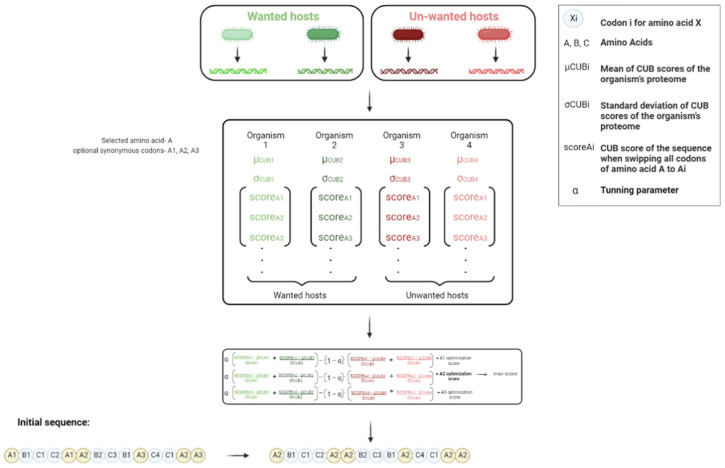
Translation efficiency selective enhancement, describing one hill climbing iteration of the translation optimization algorithm. The first step is to define the wanted and un-wanted hosts. The second step is to calculate the CUB score of each organism for all codons of amino acid A to Ai ($score_{Ai}$) and then calculate the mean ($\mu_{\mathrm{CUBi}}$) and the standard deviation ($\sigma_{\mathrm{CUBi}})$ of the CUB scores. Finally, an optimization score is calculated for each synonymous codon. All the amino acid codons in the initial sequence are switched to the codon with the maximal optimization score as calculated.

**Figure 5 biology-11-01301-f005:**
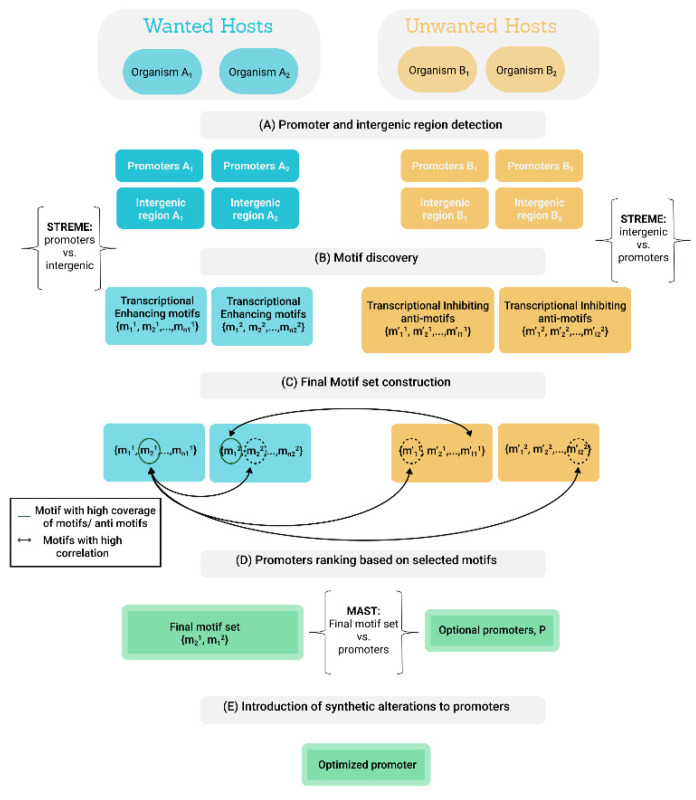
Illustration of transcription optimization algorithm. Promoter and intergenic regions sequences are extracted for every wanted and unwanted host (**A**) and are used as inputs for STREME software tool to find transcription enhancing motifs for wanted hosts and transcription anti-motifs for unwanted hosts (**B**). Transcription enhancing motifs with high correlation to other transcription enhancing motifs and/or other anti-motifs which have the highest coverage for the given microbiome population are chosen for the final motif set (**C**). Motifs in the final motif set are used to score potential candidate promoters using the MAST software tool (**D**). Synthetic promoter versions are created for top ranked promoters to further tailor the sequences based on alignment to the discovered transcription promoting motifs (**E**).

**Figure 6 biology-11-01301-f006:**
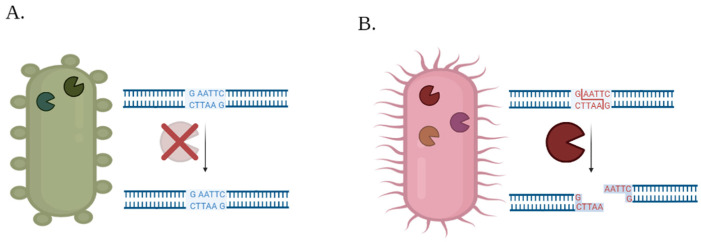
Effect of different restriction enzyme arrays in different species: (**A**) the restriction sites don’t fit to the restriction enzyme in the environment; thus, the nucleotides are not digested. In contrast, and (**B**) exhibits a pair of matching restriction sites and enzymes, so digestion occurs.

**Figure 7 biology-11-01301-f007:**
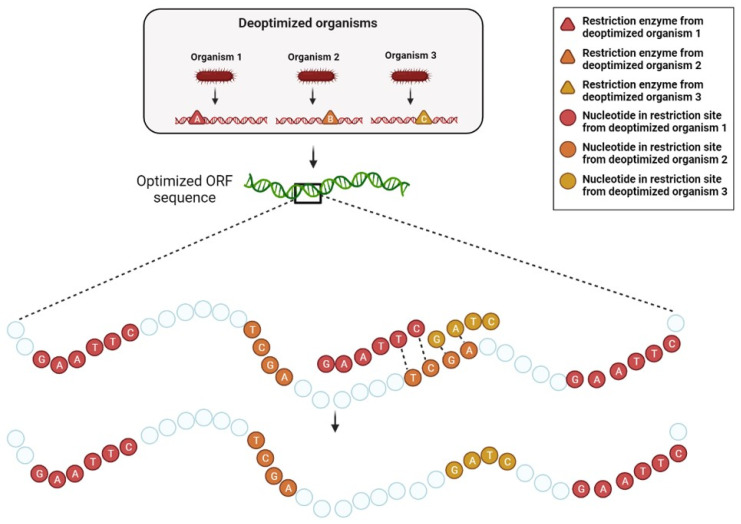
Restriction sites algorithm. The restriction enzymes (triangles) are extracted from the wanted and unwanted hosts respectively; the recognition sites of the enzymes are illustrated by squares.

**Figure 8 biology-11-01301-f008:**
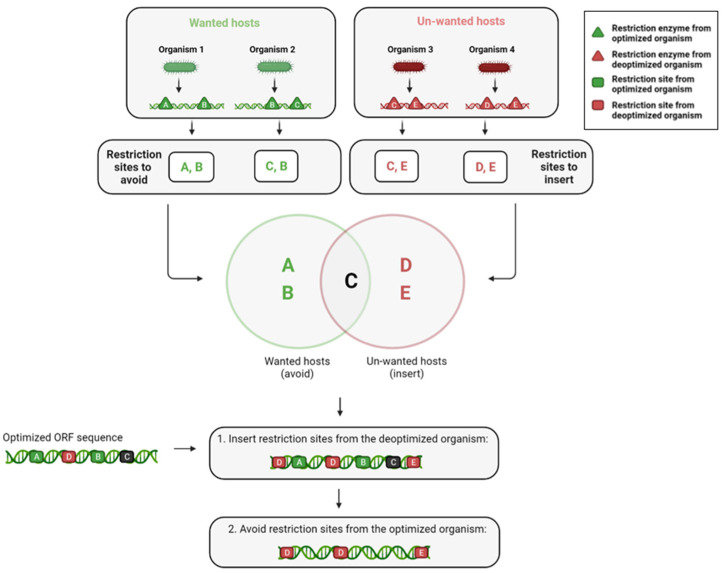
Classification of restriction sites and insertion of relevant discovered sites. The restriction enzymes (triangles) are extracted from the optimized and deoptimized organisms respectively. The selected restriction sites (squares) are the sites that contain the restriction enzymes exclusive to the deoptimized organisms. Then, the restriction sites from the deoptimized organisms are added to the sequence and the restriction sites from the optimized organisms are removed to yield the final product.

**Figure 9 biology-11-01301-f009:**
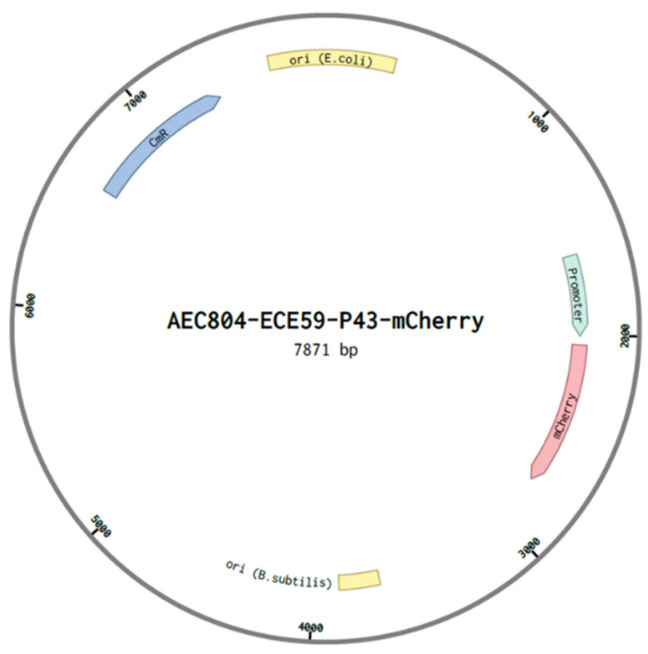
AEC804-ECE59-P43-mCherry plasmid illustration. A shuttle plasmid capable of propagate in both E. coli and B. subtilis via specific origin of replication for each of the bacteria (ORI, in yellow). The plasmid encodes for the fluorescent protein mCherry (in pink), and for Chloramphenicol resistance gene (CmR, in blue).

**Figure 10 biology-11-01301-f010:**
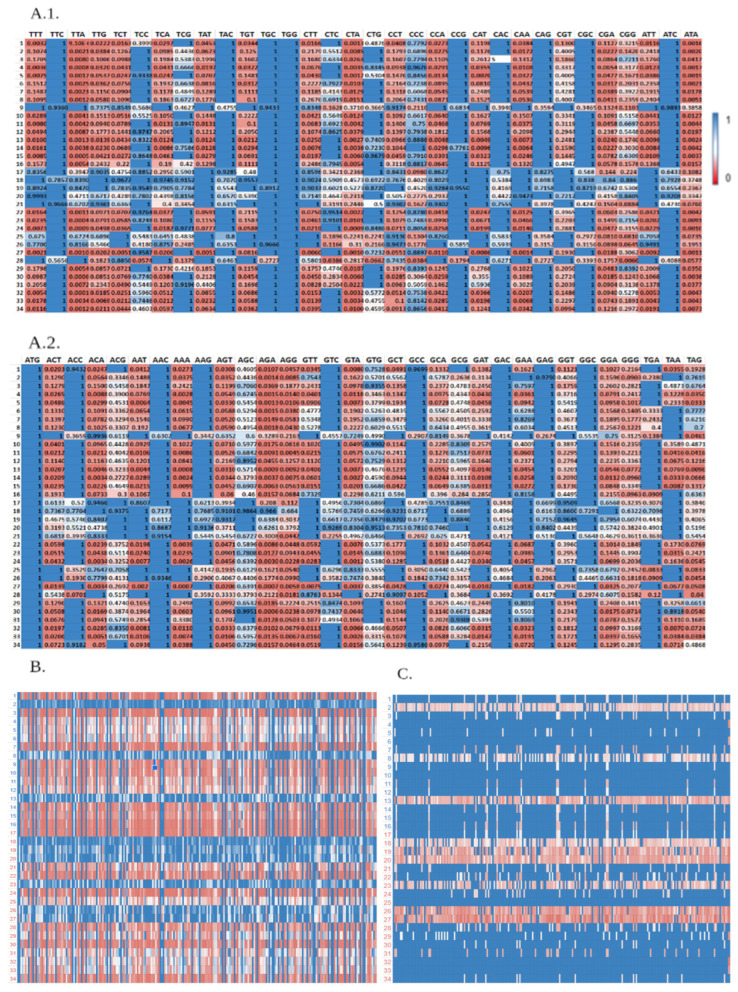
Translation efficiency optimization of the (**A.1**,**A.2**). thaliana microbiome, using the calculated CUB scores of all codons shown in figure (**A.1**,**A.2**), the initial scores of the ZorA gene in (**B**), and the final scores of the gene in **C**. The names of the organisms represented by the numbers in (**A.1**,**A.2**) and in (**B**,**C**) are detailed in Table A1 and Table A2 respectively.

**Figure 11 biology-11-01301-f011:**
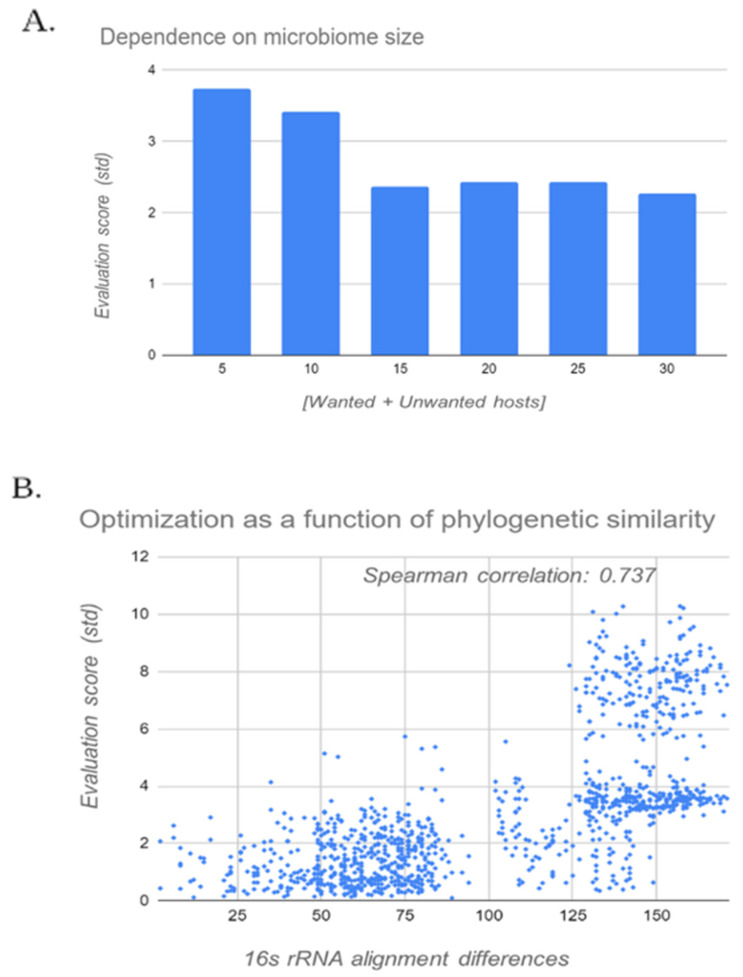
Final test of algorithm resolution and scale up: (**A**) examines 10 random splits of chosen sizes, averaged; and (**B**) shows the correlation between the performance of the model and the evolutionary distance between a pair of species (defined as the number of differences in the alignments of the 16S rRNA sequences).

**Figure 12 biology-11-01301-f012:**
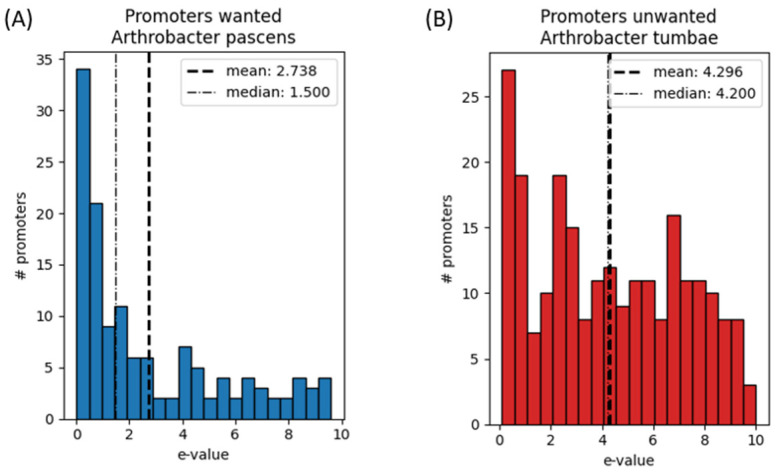
E-value scores histogram sample from a MAST run for a final motif set constructed for a pair species from the Arthobacter family, including the wanted host Arthrobacter pascens (**A**) and unwanted host Arthrobacter tumbea (**B**). In both A and B, mean and median E-values are indicated. Wanted host motifs were calculated by a STREME run using promoter sequences as primary set and intergenic regions as control set. Unwanted host anti-motifs were calculated by a STREME run using intergenic regions as primary set and promoter sequences as control set. Mean and median E-values of the wanted host are lower than mean and median E-values for the unwanted host, with a *p*-value of 7.184 × 10^−8^.

**Figure 13 biology-11-01301-f013:**
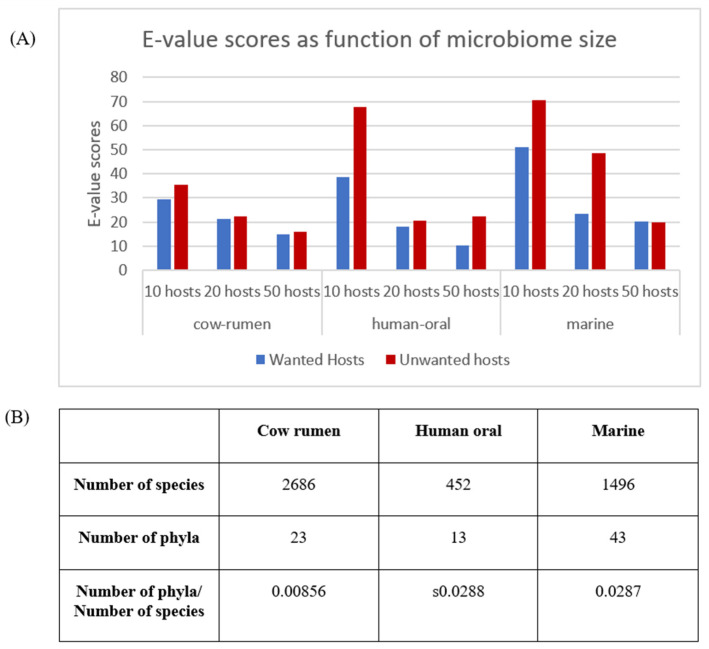
(**A**) E-value scores from a MAST run for a final motif set constructed for randomized MGnify sub-microbiomes of different sizes. The count of wanted and unwanted hosts was set to half the size of the microbiome. Only values from the 5th-percentile of the E-values calculated for the promoters of each host were considered. E-values for each group (wanted/unwanted) were calculated as the median of the median of the values of each host in the group. Test was repeated 10 times for each microbiome size; and (**B**) Meta analysis of MGnify microbiomes.

**Figure 14 biology-11-01301-f014:**
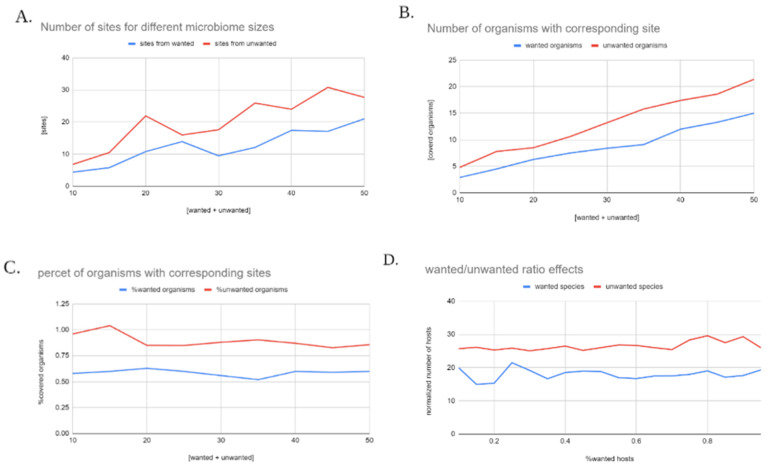
Characteristics of the engineered sequence. Random samples of 10 to 50 species were selected, and randomly split into 2 subgroups- of wanted organisms, and one of unwanted organisms. After applying the model to the defined microbiome: (**A**) exhibits the number of sites incorporated in the final sequence from each one of the two groups; (**B**) exhibits the number of organisms that have a corresponding site; (**C**) exhibits the percent of organisms that have a corresponding site; and (**D**) exhibits the normalized presence or restriction sites recognized by the wanted and unwanted hosts. For each ratio, the number of species that have a site recognized by a restriction enzyme was calculated for both groups and divided by the total number of species in the group for the sake of normalization. 30 species were randomly chosen and split into wanted and unwanted hosts according to the presented ration the exhibited results are an average of 10 runs in each condition.

**Figure 15 biology-11-01301-f015:**
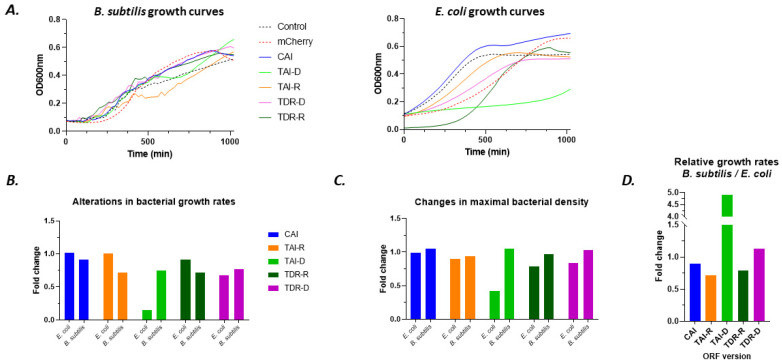
ORF modification alters the growth of deoptimized bacteria (according to optimizations in Table A1): (**A**) representative growth curves for *B. subtilis* (**left**) and *E. coli* (**right**). Control (black dashed curve) stands for bacteria containing the same plasmid backbone that lacks the mCherry gene. mCherry (red dashed curve) is the original (unmodified) version of the gene, and CAI, TAI-D, TDA-R, TDR-D, and TDR-R are modified versions of mCherry gene; (**B**) fold change in bacterial growth rates of each ORF version relative to the growth rates in mCherry; (**C**) same as in (**B**) but calculated for the average maximal density; (**D**) fold of growth rates in *B. subtilis* relative to *E. coli*.

**Figure 16 biology-11-01301-f016:**
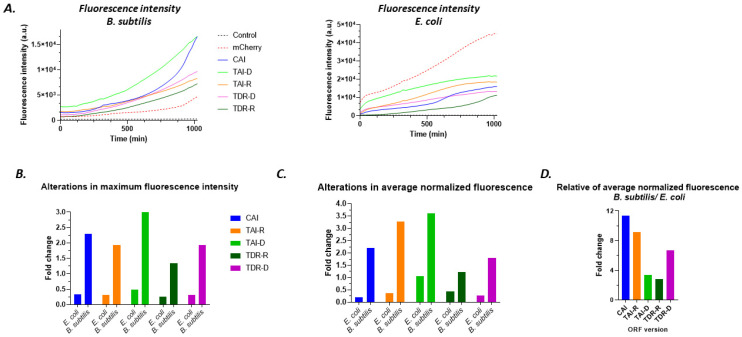
(**A**). Representative fluorescence intensity plots of all ORF variants in *B. subtilis* (**left**) and in *E. coli* (**right**). Note that the control lacked mCherry gene, and thus wasn’t exhibited fluorescence, and served for background subtraction; (**B**) fold change in average maximal fluorescence intensity of each ORF version relative to mCherry (unmodified version); (**C**) the same as in (**B**) but calculated for the average normalized fluorescence; and (**D**) fold of average normalized fluorescence in *B. subtilis* relative to *E. coli*.

## Data Availability

The sequencing data used in this study are openly available in National Center for Biotechnology Information (NCBI), BioProject number PRJNA813219. Publicly available datasets were analyzed in this study. The datasets for the translation model analysis are openly available in National Center for Biotechnology Information (NCBI), accessions: NZ_WBIN01000001, NZ_CP018863, NZ_WBJQ01000001, NZ_JAFHKT010000151, NZ_FNDT01000001, NZ_JAFBCC010000001, NZ_KV440950, NZ_CP036164, NZ_AVPK01000001, NZ_CP054795, NZ_QHKZ01000001, NZ_BCRJ01000003, NZ_SDWT01000001, NZ_PZYZ01000001, NZ_PZYY01000001, NZ_CVLG01000001, NZ_JAGGKV010000001, NZ_BILU01000001, NZ_LYPB01000049, NZ_QPJD01000001, NZ_JABMKY010000001, NZ_JACCAB010000001, NZ_RCZM01000001, NZ_LT629711, NZ_LMBV01000001, NZ_CP017704, NZ_PJNE01000001, NZ_JAEMWV010000001, NZ_VHJD01000001, NC_020541, NZ_AJXU01000028, NZ_BJYX01000001, NZ_BMNZ01000001, NZ_JACGWX010000001. The dataset used for the analysis of the restriction enzyme model is publicly available in the following link: http://rebase.neb.com/rebase/rebase.html (accessed on 1 July 2022). The datasets used for the transcription model are available in the MGnify genomes project, added in the following link: https://www.ebi.ac.uk/metagenomics/browse/genomes (accessed on 1 July 2022).

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
