# Peer review of "Modulating Gene Expression within a Microbiome Based on Computational Models"

_biology, 2022, doi:10.3390/biology11091301_

Round 1
Reviewer 2 Report
This paper developed a technique for microbiome engineering and demonstrated its computational and experimental capabilities. The method could establish a complete and automatic strategy by effectively integrating the different parts in the gene expression process. The paper is interesting and prove its performance by using both the in silico and in vitro studies to validate this engineering process. However, there are many minor points needs to be addressed before acceptance. For example, There are many spaces in several pages. The equation item # in line 254 and line 258 are the same. The item number for many equations are wrong, please recheck them. The font size of equations are not the same. There are links for several references for example [13] in page 2, [17] in page3.
